# 4D Semantic Cardiac Magnetic Resonance Image Synthesis on XCAT Anatomical Model

**Samaneh Abbasi-Sureshjani**[*1]                              S.ABBASI@TUE.NL
**Sina Amirrajab**[*1]                                        S.AMIRRAJAB@TUE.NL
[1] *Biomedical Engineering Department, Eindhoven University of Technology, Eindhoven, The Netherlands*

**Cristian Lorenz**[2]                                    CRISTIAN.LORENZ@PHILIPS.COM
**Juergen Weese**[2]                                       JUERGEN.WEESE@PHILIPS.COM
[2] *Philips Research Laboratories, Hamburg, Germany*

**Josien Pluim**[1]                                           J.PLUIM@TUE.NL
**Marcel Breeuwer** [1,3]                                   M.BREEUWER@TUE.NL
[3] *Philips Healthcare, MR R&D - Clinical Science, Best, The Netherlands*

## Abstract

We propose a hybrid controllable image generation method to synthesize anatomically meaningful 3D+t labeled Cardiac Magnetic Resonance (CMR) images. Our hybrid method takes the mechanistic 4D eXtended CArdiac Torso (XCAT) heart model as the anatomical ground truth and synthesizes CMR images via a data-driven Generative Adversarial Network (GAN). We employ the state-of-the-art SPatially Adaptive De-normalization (SPADE) technique for conditional image synthesis to preserve the semantic spatial information of ground truth anatomy. Using the parameterized motion model of the XCAT heart, we generate labels for 25 time frames of the heart for one cardiac cycle at 18 locations for the short axis view. Subsequently, realistic images are generated from these labels, with modality-specific features that are learned from real CMR image data. We demonstrate that style transfer from another cardiac image can be accomplished by using a style encoder network. Due to the flexibility of XCAT in creating new heart models, this approach can result in a realistic virtual population to address different challenges the medical image analysis research community is facing such as expensive data collection. Our proposed method has a great potential to synthesize 4D controllable CMR images with annotations and adaptable styles to be used in various supervised multi-site, multi-vendor applications in medical image analysis.

**Keywords:** 4D semantic image synthesis, cardiac magnetic resonance imaging, XCAT phantom, generative adversarial network, SPADE GAN

## 1. Introduction

**Medical image synthesis and simulation** have considerably transformed the way we develop, optimize, assess and validate new image analysis and reconstruction algorithms. They address several issues the medical research community is facing such as lack of proper,

---

* Contributed equally

annotated data, clinical privacy and sharing policy, and inefficient data acquisition costs. (Frangi et al., 2018) highlights the synergistic commonality, shared challenges, advantages and disadvantages of both (hypothesis-driven) physics-based simulation and phenomenological (data-driven) image synthesis for the medical imaging community. We can perform fully controllable experiments on the computer by mechanistic simulations grounded on implementing principles of physics-based medical imaging algorithms and benefiting from defined computerized anatomical and physiological human body models. Without doubt, an accurate in-silico human anatomy plays a crucial role in this approach. The well-known four-dimensional (4D) eXtended CArdiac Torso (XCAT) (Segars et al., 2010) computerized whole body models are arguably one of the most comprehensive digital models covering a vast series of phantoms of varying ages from newborn to adult, each comprising parameterised models for cardiac and respiratory motion (Segars et al., 2013).

More recently, by increasing the availability of big data combined with both computational powers and artificial intelligence breakthroughs, phenomenological data-driven synthetic methods for generating data have grown exponentially. Significant improvements in Generative Adversarial Networks (GANs) (Goodfellow et al., 2014) have addressed the challenge of synthesizing images with realistic and coherent spatial and non-spatial properties (Donahue and Simonyan, 2019; Park et al., 2019). However, the applications of synthetic images are still limited, because the synthetic data (sampled from learned distributions) are often limited by the number and quality of existing datasets. Limited anatomically meaningful annotated images makes it difficult to generate high dimensional data reflecting both motion and volumetric changes.

In this paper, we propose a hybrid approach to bridge the gap between simulated and real datasets by mapping the real image appearance to mechanistic controllable anatomical ground truth via a data-driven generative model. We synthesize 3D+t controlled Cardiac Magnetic Resonance (CMR) images using XCAT heart model. The accurate underlying anatomical model (what we call *true ground truth*) is preserved while modality-specific texture and style are transferred from real images. This approach makes it possible to transfer the information from any domain i.e., image modality or vendor to its corresponding anatomical model and create realistic labeled sets to be used in various supervised applications. To the best of our knowledge, this is the first time to synthesize 4D semantically and anatomically meaningful images with controllable ground truths, which is of great importance to tackle the issue of limited labeled data for developing deep learning methods serving the medical image community.

## 2. Related Work

**Data-driven image synthesis** by GANs has had significant improvements in computer vision lately. In conditional image synthesis approaches some certain input data is used as the input of the generator to provide more semantic information for the image generation (Huang et al., 2018; Lee et al., 2019; Wang et al., 2018; Park et al., 2019). However, one of the challenges is that the semantic information and spatial relations of different classes might get removed in the stacks of convolution, normalization and non-linearity layers. The state-of-the-art conditional GAN by (Park et al., 2019) deploys the segmentation

masks in novel SPatially-Adaptive (DE)normalization layers (SPADE) which despite other normalization techniques, prevents the loss of semantic information.

Recent image synthesis approaches in the medical imaging community mainly focus on the idea of disentangling the spatial anatomical information (often called as content) from the non-spatial modality-specific features (called as style). For instance, the works by (Chen et al., 2019; Ma et al., 2019) proposed to mix the contents of a known domain (with available segmentation masks) with the styles learned from a new domain. These new labeled synthetic images can help in adapting the segmentation networks to the new domain. The style is either learned by a style encoder in a Variational Auto-Encoders (VAE) (Kingma and Welling, 2013) setup or is manipulated via normalization layers affecting the statistics of the high-level image representations (Gatys et al., 2016). Other recent works such as (Chartsias et al., 2018, 2019) proposed to factorize images into spatial anatomical and non-spatial modality representations by latent space factorization relying on the cycle-consistency principle. The anatomical factor is then used in a segmentation task. All these methods rely on existing labeled sets which are both limited and not controllable. Recently, (Joyce and Kozerke, 2019) proposed to use unlabeled images by learning an anatomical model in a factorized representation learning setting. Even though the segmentation masks are not needed anymore, but still their learned multi-tissue anatomical model is not physiologically accurate and does not match actual organs.

**Physics-based image simulation** can produce controllable images by combining the modality-specific principle of image formation with a rich anatomical model. The image contrast is governed by known equations and can be altered by changing a set of parameters. These parameters are known as sequence parameters specific to imaging modality protocol that in combination with tissue-specific properties can generate image contrast. In this branch of methods, (Tobon-Gomez et al., 2011) and (Wissmann et al., 2014) investigate two types of approaches based on XCAT phantom to simulate cardiac MR images. The image contrast for the first one is calculated using a numerical Bloch solver (Kwan et al., 1999) and the latter one benefits from analytical solution for Bloch equations available for cardiac cine sequence protocol. Despite having lots of flexibility and control over the image generation process, simulated images are still far from desired realism in terms of global image appearance, tissue texture, image artifact, and surrounding organs. Furthermore, in order to create a visually familiar image appearance, large scale optimization sequence-specific and tissue-specific parameters are required. These limitations have hindered the progress of using simulated cardiac images for medical imaging applications.

Taking advantage of the biophysical motion model of the heart, the second branch of the simulation method generates more realistic images by warping already existing real images. This model-based image simulation highly depends on matching the time series of cardiac data to an electromechanical heart model (Prakosa et al., 2012). This method relies on registration in which a real cardiac image is first segmented, and then deformed and warped according to the used motion model to generate a set of transformed time series of images. Differences in the motion estimated from real images and the simulated motion of the heart during warping procedure can produce registration errors. Although much of the problems are solved in the new pipeline introduced by (Duchateau et al., 2017), this warping approach is bounded by the used images and could not generate new appearances with variable contrast, surroundings and texture.

The main contribution of this paper lies in efficiently combining the controllable physics-driven XCAT anatomical model (Segars et al., 2010) with data-driven SPADE-GAN model (Park et al., 2019) in order to synthesize realistic-looking cardiac MR images. These images do not require expert annotation since the labels derived from the XCAT model serve as the ground truth segmentation map for the generated images. The spatial information provided by XCAT model are anatomically and physiologically plausible which enables the resulting images to be useful for the purpose of data augmentation. The ability to control both anatomical representation and style in cardiac image synthesis is considered as one of the main advantage of our proposed technique compared to previous techniques.

## 3. Methodology

An overview of our method is shown in Figure 1[1]. Our conditional image synthesis network is trained on real image data with their corresponding segmentation labels. We make use of the SPADE technique to preserve the anatomical content of the labels during image generation. At the inference time, we swap the used segmentation labels with our voxelized labels which are derived from the XCAT surface-based heart model. We use the flexibility of the XCAT motion model to make a set of 3D+t labels of the heart including only the classes provided by the real data. These new controlled labels are then used to synthesize new images. The details of conditional image synthesis network, image data for training and controllable 4D heart labels for inference are explained in the following.

**Conditional image synthesis** in this work is based on the method proposed by (Park et al., 2019), which we call SPADE GAN. The architecture of the generator consists of a series of the residual blocks with SPADE normalization, followed by nearest neighbor upsampling layers. During the normalization step, the layer activations are initially normalized to zero mean and unit standard deviation in a channel-wise manner and then modulated with a learned scale and bias, which depend on the input segmentation mask and vary with respect to the location. The learned modulation parameters encode enough information about the label layout and are used in different resolutions across the generator. Therefore, they avoid the wash out of semantic information which often happens with other normalization layers such as instance normalization (IN). We also used the combination of an image encoder and the generator, and replaced the input noise with the encoded latent vector to form a VAE setup. We altered the architecture of the encoder compared to (Park et al., 2019) by removing the IN layers. The encoder with IN is in charge of capturing only the global appearance of its input image, but by removing IN we allow the spatial information to be transferred as well. Then the generator's task is to combine the encoded (global and local) style and the content coming from the semantic segmentation mask to synthesize an image. This setup is useful in controlling the style of synthetic images and the reconstruction of the surrounding organs of the heart. The architecture of the discriminator, the losses and training settings are kept unchanged.

**The real dataset** used for training the network is the Automated Cardiac Diagnosis Challenge (ACDC) dataset (Bernard et al., 2018). This dataset consists of Cine MR images of 100 patients. The spatial resolution goes from 1.37 to 1.68 $mm^2/pixel$ and images cover the cardiac cycle completely or partially. In total, there are 100 end-systolic and

---

1. An animated version of our methodology is available here: https://bit.ly/2Ggr61j

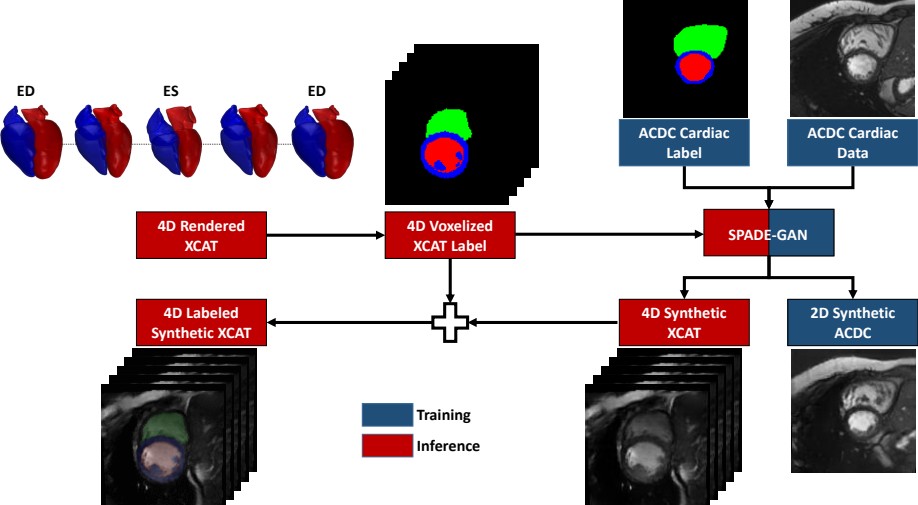

Figure 1: An overview of our method. In training (blue blocks), we use the ACDC images with their corresponding segmentation masks as inputs of the SPADE GAN. At inference (red blocks), we substitute the ACDC labels with our 4D voxelized XCAT labels created from the XCAT heart surface model to synthesize new images (4D synthetic XCAT). The rendered version for the XCAT heart surface model is shown for five time frames. The 4D voxelized XCAT labels cover heart from apex through mid to base location for one cardiac cycle. The same labels are used as the ground truth for the new synthetic images (4D labeled synthetic XCAT).

100 end-diastolic phase instances, with an average of 9 slices. The segmentation masks for left ventricle (LV) blood pool, LV myocardium, and right ventricle (RV) blood pool are available. We pre-process the data by subsampling them to $1.3 \times 1.3$ $mm$ in-plane resolution (fixed inter-slice resolution) and take a central crop of the images with $128 \times 128$ pixels. All the intensity values are scaled between -1 and 1. The SPADE GAN is trained on the entire 2D set of image-mask pairs of this dataset for 100 iterations, using Adam optimizer with learning rate if 0.0002, batch size of 32 on 2 NVIDIA TITAN Xp GPUs. We use the VAE setting with larger images ($256 \times 256$) for a better demonstration.

**Controllable 4D heart model** is the key element of our method. We employ the 3D+t NURBS-based surfaces of the XCAT heart model which is anatomically based on 4D cardiac-gated multislice CT data and its motion model is parameterized by tagged MRI data. To create an accurate 4D voxelized heart model, the XCAT program offers various parameters to control morphological (heart shape) and physiological (heart motion) features of the heart. These parameters include heart scaling factors in 3D; the length of the beating heart cycle; left ventricle volume at end-diastole, end-systole, and three intermediate phases; cardiac cycle timing which is the duration between different phases.

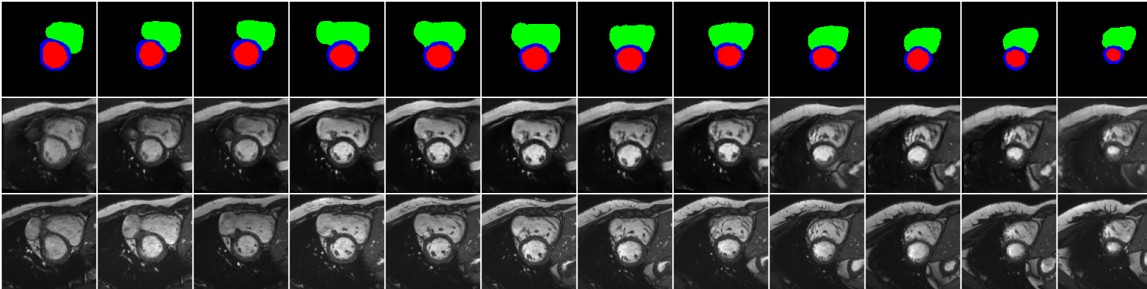

Figure 2: The synthetic ACDC slices from apex to base location for one subject of the ACDC dataset at the end-diastolic phase. The rows from top to bottom show the input label maps, the synthetic and real images respectively.

We keep the geometrical scaling of the XCAT heart unchanged, set the length of beating heart cycle to 1 $sec$ (60 $heartbeats/min$) and output 25 time frames along one heart cycle. Voxelization of surfaces can be done at any desired resolution. We create 1 $mm$ isotropic in-plane resolution for 18 slices perpendicular to the long axis of the heart to form the short axis view of the heart which shows the cross-section of the left and right ventricles.

Our main contribution comes at the inference time. We use our 4D voxelized XCAT labels (sets of 2D slices at different locations and times) as the inputs of the generator and synthesize their corresponding realistic images. The synthetic slices reflect the accurate anatomical model with modality-specific texture and style. These new images together with the true ground truth create a new 4D synthetic XCAT dataset, which can be used in various applications. Results are presented and discussed in the next sections.

## 4. Results

First, we show the synthetic images when using the labels of the ACDC dataset as inputs of SPADE GAN. Figure 2 shows different synthetic slices (from apex to base) for one subject of the ACDC dataset in the end-diastolic phase. Similar results for the end-systolic phase are depicted in A, Figure 5. As seen in these figures, the synthetic images are coherent between slices even though the training is done on 2D slices. Moreover, the three classes of interest in the heart have been reconstructed reasonably well. There are some differences between the background tissues in real and synthetic images. This is because all different tissues in that region are mapped into one class in the label map (background shown by black in the label map). Thus the SPADE GAN is not able to preserve their spatial information.

The main results, which are the synthetic images corresponding to the XCAT labels are shown in Figure 3. For visualization purposes, we fix the location and vary the time frame. The results for 12 time frames from end-diastolic to end-systolic phase (from left to right) are shown at the base location of the short axis view of the heart. Due to limited space, similar results for other time frames and locations are shown in A, Figure 6 and Figure 7. Additionally, a 4D visualization of our results is available here: https://bit.ly/2REVAzB.

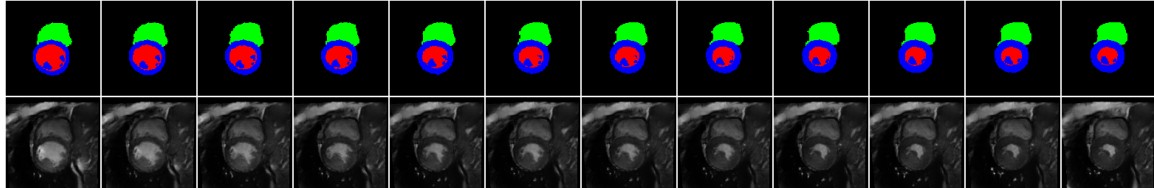

Figure 3: 4D synthetic images on XCAT labels for 12 time frames from end-diastolic to end-systolic phase (left to right) are shown at the base location of the short axis view of the heart. The rows represent the input label maps and their corresponding synthetic images.

As seen in these figures, for a fixed location, the classes of interest are generated according to the input label map, while the background is consistent and coherent.

In another experiment, we test our modified VAE setup on the 4D voxelized XCAT labels to show the capability of the method in generating synthetic images in which the global and local styles are matched to images from an unseen dataset. Some sample results are shown in Figure 4. The input images of the encoder (representing the style) are depicted in the first column. Two different synthetic images for each style are shown in the second and third columns, and the label maps (the inputs of the SPADE layers) are shown on the top left corner of the resulting synthetic images. In these images the local and global appearance of the style images are transferred to the synthetic images, while keeping the classes of interest intact. This VAE setup provides an additional control on our image generation. The generator is capable of creating realistic heart models, while the encoder transfers the information related to the other surrounding organs. For the sake of comparison, using the same combination of style and label maps, the resulting synthetic images when the IN layers are kept in the style encoder are also shown in the fourth and fifth columns. In these cases, only the global style is transferred and the control on the surrounding regions of the heart is very limited.

## 5. Discussion and Conclusion

In this paper, we have proposed a hybrid method to use the voxelized 3D+t NURBS-based surfaces of the XCAT heart model in a deep generative network and synthesize semantically and anatomically meaningful 4D realistic CMR images with controllable ground truth labels. Even though the SPADE GAN is trained on 2D images, the synthetic images are very coherent across the other two dimensions of the labels (slice and time). Specifically, the heart that is our main focus in this work, is synthesized consistently. However, small variation and inconsistency in the background can occur because all tissues that are not of interest (i.e. not belonging to the heart) are assigned to the background class. This may be ignored when the application of the synthetic data is heart cavity segmentation. For multi-organ segmentation applications, the main limitation comes from the limited number of classes in the ACDC dataset as various organs are mapped to the background class.

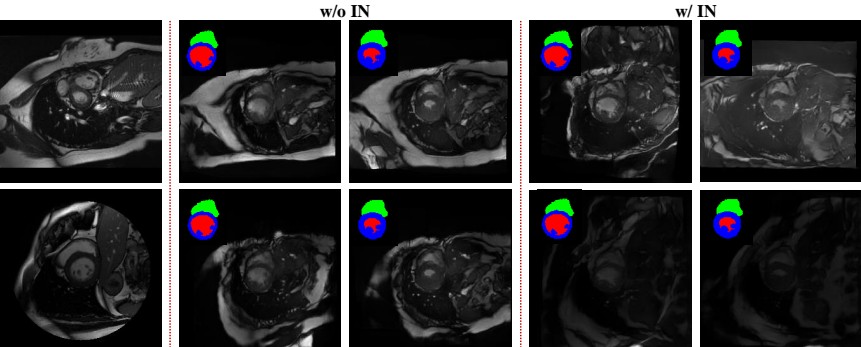

Figure 4: Transferring desired styles to synthetic XCAT images. The first column represents the desired style images. The resulting synthetic images for each style without and with IN layers are shown in the second to fifth columns. The corresponding input label maps are shown in the top left corner of the synthetic images.

Since the background label does not contain any spatial information, we only have limited control over the generated background regions through our modified VAE setting. Our style encoder encodes the local semantic information of the input style image, in addition to global style information, to a latent vector. Removing the IN layers prevents the removal of semantic information and helps in generating consistent background for nearby slices. Definitely, multi-tissue or multi-class segmentation of background can help in generating more realistic results as it provides more information to the generator. Moreover, using other MR modalities such as T1-weighted and late gadolinium enhancement extends the variations in the global style compared to the limited styles learned from the ACDC dataset with cine MR contrast. It is worth mentioning that for the 4D voxelized XCAT labels, we only selected the classes matching the labels of the ACDC dataset. If we use another dataset with more labels, we can use more classes of the XCAT model as well.

The main advantage of using the XCAT model is that not only it can be controlled and modified to generate new heart labels, it can also provide anatomically meaningful accurate ground truth for different time frames. So the 4D labeled synthetic CMR images can potentially be employed in cardiac supervised tasks. This is a great advantage over the previous approach by (Joyce and Kozerke, 2019) in which their estimated mutli-tissue segmentation map is not necessarily anatomically plausible. Moreover, their deformable model does not provide physiologically meaningful information since its motion is modelled by an interpolation in the latent space between anatomical shapes of end-systolic and end-diastolic phases.

Our future works are twofold: i) improving the control over generating the background by dividing it into an approximated multi-organ segmentation map which eventually results in more temporary consistent background and ii) quantitative application-based evaluation of the synthetic images by deploying them in a heart segmentation task for multi-site, multi-vendor scenarios. We use our proposed approach to generate a large virtual population with

various anatomical and style variations and utilize the synthetic images in different data augmentation strategies for the cardiac cavity segmentation task. The goal is to investigate the utility of the synthetic data in training deep learning algorithm for segmentation and evaluate that the data generated by this approach is clinically meaningful to replace the need for real data.

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

## Appendix A. Additional Figures

This section includes additional synthetic images. Figure 5 includes synthetic slices for the fixed end-systolic phase for one patient of the ACDC dataset.

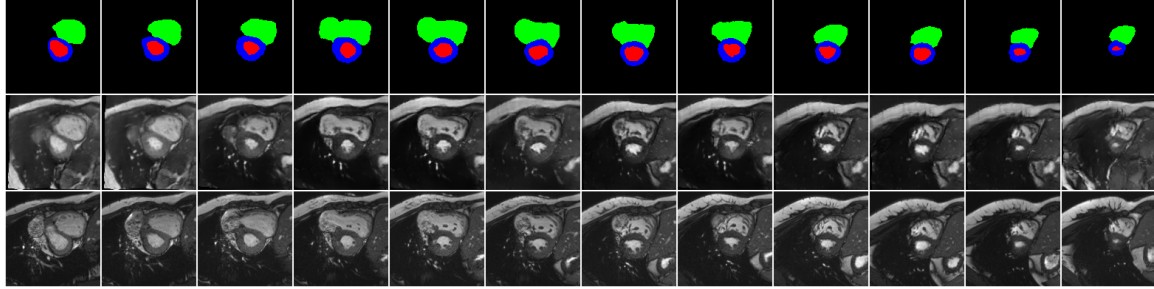

Figure 5: The synthetic ACDC slices from apex to base location for one subject of the ACDC dataset at the end-systolic phase. The rows show the input label maps, the synthetic and real images respectively

Figure 6 shows the generated samples for XCAT labels for 12 time frames from end-diastolic to end-systolic phase while fixing the location. Figure 6(a), 6(b) correspond to apex and middle locations respectively. Similarly, the results for end-systolic to end-diastolic phases, corresponding to apex, middle and base locations are shown in Figure 7.

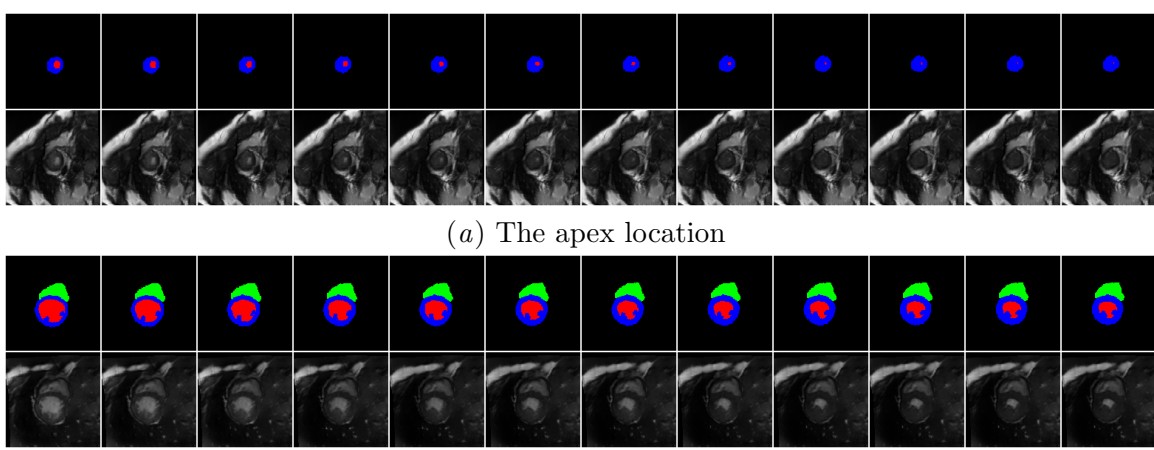

(a) The apex location

(b) The mid location

Figure 6: 4D synthetic images on XCAT labels for 12 time frames from end-diastolic to end-systolic phase at apex and mid locations of the short axis view of the heart. In each figure, the first and second rows represent the input label map and their corresponding synthetic images respectively.

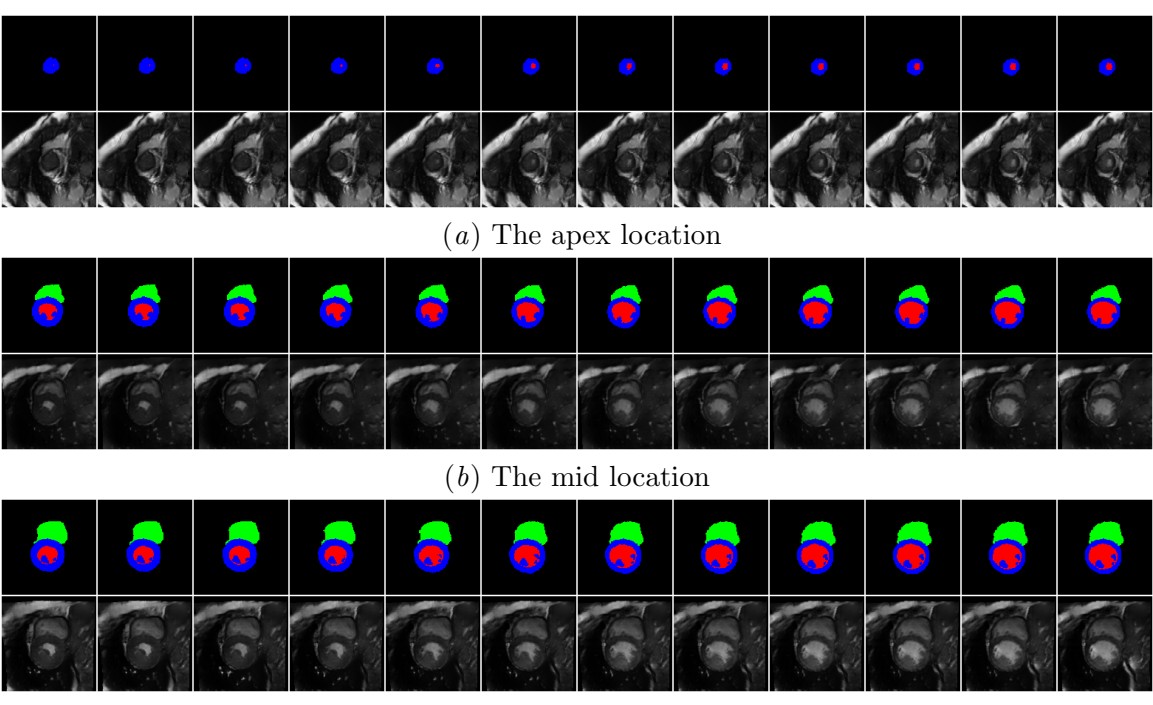

(a) The apex location

(b) The mid location

(c) The base location

Figure 7: 4D synthetic images on XCAT labels for 12 time frames from end-systolic to end-diastolic phase at three different locations of the short axis view of the heart. The first and second rows represent the input label map and their corresponding synthetic images respectively.

