# OpenReview forum: "4D Semantic Cardiac Magnetic Resonance Image Synthesis on XCAT Anatomical Model"
_MIDL.io/2020/Conference — MIDL 2020_

### Official Review · AnonReviewer4 · 2020-03-12
**generation of 4D images with anatomical structures ensured while some clarification and details are needed**

**Rating:** 2
**Confidence:** 4
**Recommendation:** Poster

**Summary:**

The proposed a hybrid method based on SPADE for cardiac MR images synthesis conditioned on segmentation labels obtained from anatomical structure of a physical cardiac model. A SPADE-like model is trained on ADCD dataset A controllable 4D model is used to generate segmentation labels similar to those of ACDC in a 3D+t manner, those labels are then used for 4D generation. By using this controllable 4D model, the proposed method ensures the realism of the anatomical structures in the generated images.

**Strengths:**

The paper is well-structured and provides details of architectures and generation visualization. Besides the effort on generating realistic image, the method also considers the correctness of anatomical structures, which is critical for down-stream applications.  In the presented results and animation, the quality of generation is visually good. It would be interesting to see if the generated images by the proposed method can be used to improve segmentation of cardiac images or images of other organs.

**Weaknesses:**

Some improvements may be needed:

1) although the visualized results show good quality of generation, qualitative measures are also needed to evaluate the generation quality, such as FID, PSNR etc, and compare to the baseline methods.
2) the generated data is a 3D+t format, but it seems the method does not explicitly enforce temporal consistency in the generated images.

**Justification Of Rating:**

The proposed method describes a framework to generate 4D cardiac MR images using labels obtained from physical models. The framework is based on SPADE-GAN and extends to 4D generation. The main advantage is that the meanings of anatomical structures are ensured and the generation quality is satisfactory. However, evaluation may be necessary to compare this method to the existing methods in order to better understand its advantage. More details should be provided to enable reproducibility. On the other hand, the weakness is that, the method does not seem to explicit ensure temporal consistency although this may be ensured by the physical model, some discussion/clarification on this may be needed.

**Paper Type:**

methodological development

**Questions To Address In The Rebuttal:**

The authors may consider adding the follow details and discussion:

1) comparison with other related methods, and possibly also qualitative evaluation of those methods;
2) details of the SPADE-GAN architecture used in this paper and the loss function used for training
3) apart from cardiac images, how difficult may it be as to extend the method to the generation of other organs?



**Special Issue:**

no

---

> ### Author Response · Authors · 2020-03-27
> **We agree with the suggestions of the reviewer and addressed the comments point-by-point.**
>
> Thank you for your comments and suggestions.
>
> > Comparison with other related methods, and possibly also qualitative evaluation of those methods
> We addressed your comments regarding the evaluation of the synthetic images in the official comments of the paper.
>
> > Details of the SPADE-GAN architecture used in this paper and the loss function used for training
> We agree with your suggestion about adding extra explanations about the architecture of the SPADE GAN and training losses. We only focused on the key concept of the SPADE architecture (Park et al., 2019) which is utilizing the SPADE layers for image synthesis. The main difference of the SPADE GAN image generation technique and the previous architectures (pix2pixHD (Wang et al, 2018)) is employing the SPADE layers in order to control over the semantic content transferred to the synthetic images. We did not add further explanation about the image synthesis since it is not the contribution of this paper and due to limited space.
>
> > Apart from cardiac images, how difficult may it be as to extend the method to the generation of other organs?
> Thanks for pointing out the extension to other organs. Since the XCAT phantom provides ground truth models for other organs in the body as well, our proposed image generation approach is extendible to other organs as long as a properly labeled real dataset is available for training the network. We trained on the ACDC dataset with limited annotations of the heart regions. As seen in the results, the image generation is consistent for the heart tissues because there is enough guidance for image generation during the training. If the annotations of other organs are also available, it is possible to train the SPADE GAN using them and use the corresponding labels of the XCAT model during the inference time.
>
>
> * Ting-Chun Wang, Ming-Yu Liu, Jun-Yan Zhu, Andrew Tao, Jan Kautz, and Bryan Catan-zaro. High-resolution image synthesis and semantic manipulation with conditional GANs.In2018 IEEE/CVF Conference on Computer Vision and Pattern Recognition, pages 8798–8807, June 2018.

---

### Official Review · AnonReviewer2 · 2020-03-13
**beautiful images but poorly written**

**Rating:** 2
**Confidence:** 4

**Summary:**

This paper applied SPADE-GAN for 4D cardiac MRI synthesis. Personally, I donot get the main ideas of this paper, it seems to me it is applying a technique (SPADE-GAN ) to a specific task (Cardiac MRI synthesis).  The experiments are not well done at all. There is even no quantitative results or analysis. We cannot make a conclusion just based on visualization. As for significance, I think it should be further explored if the synthetic data are meaningful or not for clinic. I think this paper should be further improved before publication.

**Strengths:**

1. The problem, i.e., 4D MRI synthesis is interesting. Though many synthesis tasks exist in medical image synthesis field, they are limited in 2D/3D.
2. The visualized images are beautiful. There are many well presented images in the appendix and also including a 4d MRI in the dropbox. It is really cool.

**Weaknesses:**

1. This paper is really poorly written, very hard to follow. Sometimes, they are even inconsistent, for example, what's the contribution of this paper.
2. The novelty of this paper is limited, to me, it is applying a SOTA technique on a medical image synthesis task. If it is a pure application paper, then I'd like to read an entire story.
3. There are no quantitative results or analysis.
4. How to make sure if the results are reliable or not.
5. Many medical image synthesis tasks are not mentioned. The authors mainly introduce work from natural image field.

**Justification Of Rating:**

1. The paper is poorly written and hard to follow with limited novelty.
2. The experiments are not well done at all. There are even no quantitative analysis (We cannot make a conclusion just based on visualization.).
3. Though the topic is interesting, the novelty is limited.

**Paper Type:**

validation/application paper

**Questions To Address In The Rebuttal:**

1. How to validate if the synthetic MRI are clinic meaningful? Do you have any plan?
2. Quantitative results/analysis.

**Special Issue:**

no

---

> ### Author Response · Authors · 2020-03-27
> **We agree with the suggestions of the reviewer and addressed the comments point-by-point.**
>
> Thank you for your comments and suggestions.
>
> > The contribution of the paper and evaluation of the results
> We addressed the comments regarding the contribution of the paper and the evaluation of the synthetic images in the official comments of the paper.
> We would like to emphasize that the novelty of the paper is not the image generation architecture (SPADE GAN), but is the efficient integration of physics-driven anatomical models and data-driven image synthesis approach. To the best of our knowledge, this is the first time to introduce new and accurate anatomical variations in image synthesis via anatomical phantoms. This work is a portion of a larger pipeline for tackling the limited labeled data in the clinic and The usefulness of the synthetic images will be investigated quantitatively in a supervised cardiac cavity segmentation task as a future work.
>
>
> > How to validate if the synthetic MRI are clinic meaningful? Do you have any plan?
> We addressed this question in the general comments to all reviewers
>
> > Quantitative results/analysis.
> We addressed this question in the general comments to all reviewers.

---

### Official Review · AnonReviewer1 · 2020-03-14
**4D cardiac image synthesis**

**Rating:** 3
**Confidence:** 4
**Recommendation:** Poster

**Summary:**

This paper provides a method for 4D cardiac MR image synthesis. The model takes the XCAT heart model as the ground truth and employs SPADE GAN for conditional image synthesis. Then a style transfer network is used to adjust the style. The method could generate realistic and controllable 4D cardiac MR images.

**Strengths:**

- The proposed method has generated controllable and realistic images in 4D. Synthesis image is a great way to combat the limited training data in medical images.
- The proposed method is based on one recent publication on CVPR 2019, SPADE GAN, which is a spatially constrained method for image synthesis.
- A well-known phantom model is used for image synthesis.
- Related work are thoroughly cited and discussed.

**Weaknesses:**

- The background in synthetic cardiac images could not be controlled, and are not consistent spatially and temporarily.
- A brief description of the SPADE GAN would help the understanding and completeness of the manuscript. The writing could be more organized.
- There is no quantitative analysis.

**Justification Of Rating:**

This paper provides a method for generating realistic 4D cardiac images. The visual results are very promising with the ground truth region controlled. Lots of followup work could be inspired by this work, so I suggest acceptance of this paper.

**Paper Type:**

both

**Questions To Address In The Rebuttal:**

It is not very clear to me how the VAE setting is helpful to control the style.

**Special Issue:**

no

---

> ### Author Response · Authors · 2020-03-27
> **We agree with the suggestions of the reviewer and addressed the comments.**
>
> Thanks for your suggestions and comments.
>
> >  It is not very clear to me how the VAE setting is helpful to control the style.
>  In the general comments to all reviewers, we addressed your questions about the VAE setting and the quantitative analysis.

---

### Official Review · AnonReviewer3 · 2020-03-18
**Promising new approach for generating 4D cardiac image data that may greatly benefit the wider community**

**Rating:** 3
**Confidence:** 4
**Recommendation:** Poster

**Summary:**

The authors present a method for generating realistic, computational physics-driven 4D images of cardiac MRI. They use the XCAT model of the heart, generating segmentation labels at 25 cardiac phases from a physics-driven deformation of a biventricular heart model across the cardiac cycle. Cardiac Cine MRI data of 100 patients with matching biventricular labels are used from the AC/DC Challenge. Using the recently proposed SPADE GAN, a model is trained to generate synthetic cardiac Cine MR images conditioned on the segmentations either from the labels of the MRI data or from the segmentations generated from the XCAT model. At inference, synthetic MR images can be generated for a given set of labels, where the anatomical information encoded by the segmentations is preserved in the generational images. Furthermore, the specific style from a given MR image can be transferred to a generated image while producing anatomy that is consistent with a target segmentation.

**Strengths:**

- The paper is well written and clearly motivated.
- The paper provides an elegant bridge between highly-controllable, physics-driven model of cardiac deformation and the challenging domain of realistic medical image generation, helping to address the common challenge of limited labeled training data for applications such as segmentation and biometric quantification of cardiac image data.
- The ability to control anatomical and physiological parameters to produce a wide variety of realistic segmentations across the cardiac cycle, and then to generate realistic MR data from these segmentations holds great potential for training task-specific deep learning models with limited data.

**Weaknesses:**

- Although the overall idea is novel, the architecture novelties are very limited.
- The authors explain that IN layers are removed from the encoder described in the SPADE paper for the VAE model. They do not show results demonstrating the value of this.
- There is a lack of quantification of performance metrics.

**Detailed Comments:**

- `Bibliography entry #1 does not contain an author list.
- Citation 'Bernard et al.' in text does not have a date.
- Page 7: 'SADE' should be 'SPADE'

**Justification Of Rating:**

The authors demonstrate the power of combining physics-driven 4D simulation of cardiac deformation with recent developments in conditional image generation using SPADE to generate physiologically realistic cardiac MRI sequences which are anatomically consistent with a provided segmentation produced from the XCAT model. This holds great potential for generating training data in limited labeled data settings.

**Paper Type:**

both

**Questions To Address In The Rebuttal:**

- What were the observed differences of keeping and removing IN layers from the encoder of the VAE?
- indeed the background is not temporally consistent, and in the case of the VAE examples, does not look very realistic. The authors have stated this will be addressed in future work however.
- Application of the idea to a specific task would be important to demonstrate its utility.
- This idea could be very helpful for the wider community - will the code be made publicly available?

**Special Issue:**

no

---

> ### Author Response · Authors · 2020-03-27
> **We agree with the suggestions of the reviewer and addressed the comments point-by-point.**
>
> We thank the reviewer for the thoughtful comments.
>
> >  What were the observed differences of keeping and removing IN layers from the encoder of the VAE?
> We addressed your question in the general comments to all reviewers.
>
> >  Indeed the background is not temporally consistent, and in the case of the VAE examples, does not look very realistic. The authors have stated this will be addressed in future work however.
> We agree with your comment. The main reason is that the labels for the surrounding regions of the heart are not available, so there is no guidance (i.e. label mask) for the SPADE-GAN network for generating other organs than the heart. Since only the labels of the heart classes are connected temporally, the synthetic images are consistent at the heart region, but not the surrounding regions. We will address this in the future work by providing more guidance to the network using more labels especially for the organs visible in the background class.
>
> > Application of the idea to a specific task would be important to demonstrate its utility. - This idea could be very helpful for the wider community
> We totally agree with your suggestions.  We addressed your comments regarding investigating the usefulness of synthetic images in the official comments of the paper.
>
> > will the code be made publicly available?
> Since our method is based on the SPADE method (Park et al., 2019), a great part of the code is already publicly available (https://github.com/NVlabs/SPADE). We will make the rest of the codes publicly available to benefit the research community.
>
> > Detailed Comments:
> We accordingly added the missing parts in the bibliography and modified the typo that you pointed out.

---

### Author Response · Authors · 2020-03-27
**General comments regarding the contributions of the paper, evaluation of synthetic images and the VAE setup**

We appreciate your valuable feedback on our manuscript. Here, we address the main questions asked by you.

> Main contributions
The main contribution of this paper lies in efficiently combining the controllable physics-driven XCAT anatomical model (Segars et al., 2010)  with data-driven SPADE-GAN for image synthesis (Park et al., 2019). The anatomical variations introduced in the literature via VAEs or statistical shape deformations (Joyce et al., 2019, Corral Acero et al 2019) are not only limited but also not necessarily anatomically plausible. The main advantage of our proposed technique compared to these techniques is that we are able to introduce a wide range of accurate anatomical variations through the XCAT phantoms. The SPADE-GAN transfers the modality-specific MR image characteristics to the label maps and generates a new realistic-looking, anatomically plausible image without any limitation. We believe our proposed technique is of great importance to tackle the issue of limited labeled data for developing deep learning methods serving the medical image community and has a great potential to inspire other researches.

> Quantitative evaluation
We plan to use our proposed approach to generate a large virtual population with various anatomical and style variations and utilize it in different data augmentation strategies for the cardiac cavity segmentation task in multi-site, multi-vendor scenarios.  The goal is to investigate two questions: a) whether the new anatomical variations integrated in the synthetic images help in improving the segmentation results; b) whether the synthetic images are clinically meaningful to replace the real images. This helps in addressing the challenge of the limited number of labeled dataset in medical image analysis.

As for other ways to evaluate the synthetic images in the literature, the most common metric is the Fréchet Inception Distance (FID) (Heusel et al., 2017), which is often used for evaluating the synthetic photographic images (e.g. ImageNet) and not the radiological images.  It is a relative metric and it highly depends on the methodology and settings used for computing the scores, thus it is not a reliable metric to use for evaluation (Shmelkov et al., 2018).

> The VAE setup
There are several reasons for our proposed change in the VAE setup. First of all, in this work, the modality-specific features are all learned from the ACDC dataset with cine MR contrast. Thus the variations in the global style learned by the style encoder are very limited. Secondly, the background class includes various organs surrounding the heart. Since the guidance at these regions is limited to one class during the training, the network is not capable of generating consistent background tissues across the third axes and time at the inference time. We know based on the previous studies e.g. (Park et al., 2019; Huang et al, 2017; Dumoulin et al, 2016) that normalization layers play an important role in controlling the amount of semantic information transferred to the output image. Thus we propose to provide guidance for the background region, not through the label maps but via a modified style encoder that encodes the spatial semantic information in addition to global style information to a latent vector. Removing the IN layers deserves the purpose and helps in generating consistent background for nearby slices. As a future work, we aim to extend the variations in the global style by using other MR modalities such as T1-weighted and late gadolinium enhancement for training the network.

 In order to clarify these points, we propose the following modifications to the revised version of the paper. We will add extra explanations to Section 2 and 5 to clarify the contribution of the paper and future works regarding evaluating the images quantitatively. Moreover, we will demonstrate some sample results of the VAE setting when the  IN layers are included.

We look forward to hearing from you in due time regarding our submission and to respond to any further questions and comments you may have.

* Corral Acero, J., et al., SMOD - data augmentation based on statistical models of deformation to enhance segmentation in 2D cine cardiac MRI. In: Coudière, Y., Ozenne, V., et al. (eds.) Functional Imaging and Modeling of the Heart .pp. 361–369. Springer International Publishing, Cham (2019)
* Joyce, T., et al.: 3D medical image synthesis by factorised representation and deformable model learning. In: Burgos, N., et al. (eds.) Simulation and Synthesis in Medical Imaging. pp. 110–119. Springer International Publishing, Cham (2019)
* Heusel, M., et al., GANs trained by a two time-scale update rule converge to a local Nash equilibrium. In NeurIPS, pp. 6626–6637, (2017).
* Shmelkov K., et al., How Good Is My GAN?. In: Ferrari V., et al. (eds) ECCV 2018. LCS, vol 11206. Springer, Cham (2018)
* Dumoulin V, et al., A learned representation for artistic style. In ICLR, 2016.

---

### Meta-Review · Area_Chair1 · 2020-04-10
**MetaReview of Paper200 by AreaChair1**

**Rating:** 3
**Recommendation For Accepted Papers:** Oral, Poster

**Metareview:**

This paper presents work on how to combine data-driven and physics-driven trainning of deep learning algorithms to improve the quality of image synthesis with less annotations. This is a relevant topic and while the manuscript received some critiques, the authors have done a good job in responding to them and have proposed specific improvements they will do in the final manuscript. I think this paper will generate interesting discussions at MIDL and is worth presenting.

**Paper Type:**

methodological development

**Special Issue:**

no

---

### Decision · Program_Chairs · 2020-04-11

Accept